# Challenges and Potentialities of Sustainability in the Institutional Food Market of Family Farming

Edinete Rita Folle Cecconello [1], Leila Dal Moro [1], Cristian Rogério Foguesatto [2], Raquel Breichtenbach [3], Alcindo Neckel [1], Caroline Pauletto Spanhol [4], José Eustáquio Ribeiro Vieira-Filho [5] and Giana de Vargas Mores [1,*]

[1] Business School and Polytechnic School, ATITUS Educação, Passo Fundo 99070-220, Brazil; edinetececconello@gmail.com (E.R.F.C.); leila.moro@atitus.edu.br (L.D.M.); alcindo.neckel@atitus.edu.br (A.N.)

[2] Agronomy School, Federal University of Goias, Goiânia 74690-900, Brazil; cristian17agro@gmail.com

[3] Federal Institute of Rio Grande do Sul, Sertão 99170-000, Brazil; raquel.breitenbach@sertao.ifrs.edu.br

[4] Business School, Federal University of Mato Grosso do Sul, Campo Grande 79074-460, Brazil; caroline.spanhol@ufms.br

[5] Instituto de Pesquisa Econômica Aplicada (IPEA), Brasília 70390-025, Brazil; jose.vieira@ipea.gov.br

* Correspondence: giana.mores@atitus.edu.br

**Abstract:** Family farming is relevant for the generation of employment and income in the countryside. The objective of this study is to analyze the challenges and potentialities in the socioeconomic dynamics of agriculture through the institutional food market in the area of operation of a family farming cooperative. A qualitative case study was carried out in the north of the state of Rio Grande do Sul, Brazil, through eighteen in-depth interviews. Data were analyzed using the content analysis method, considering five categories: economic, social, environmental, food security, and family succession. The results indicate that labor shortage and excessive bureaucracy are constant challenges. Furthermore, effects of the COVID-19 pandemic, lack of information, and lack of succession are among the main challenges of the institutional market from the perspective of the social agents participating in the research. On the other hand, the institutional commercialization channel, via the National School Feeding Program, is a consolidated public policy in the region, and one of the greatest benefits identified was the increase in the income of the cooperative members. Among the main potentialities of the institutional market from the agents' point of view are income guarantee, institutional relations, support from cooperatives and technical assistance and rural extension, women's involvement, and productive diversification. The results can support the implementation and strengthening of similar programs in other countries.

**Keywords:** food security; public policy; agribusiness; regional development

## 1. Introduction

Since 1990, Brazilian family farmers came to be seen by the state as an important social category in agriculture, the result of demands from rural social movements, scientific research, and the socioeconomic relevance of the sector, above all due to the number of people employed and the diversity of food produced. However, in 2006, Brazil legally recognized the category through the Family Farming Law, n° 11.326/2006 [1]. Family farming differs from the others because it uses the labor of family members and explores and manages different types of food, which is the main source of income for these families. In addition to small rural producers, it involves the indigenous population, quilombola communities, and foresters [1].

After this, to reduce poverty in the rural areas, public policy instruments were created or modeled. An example of this is the National Program for Strengthening Family Farming (Pronaf), which contributed to the implementation and/or reformulation of programs in

the 2000s, especially the Food Acquisition Program (PAA) in 2003 and the National School Feeding Program (PNAE) in 2009 (FNDE, 2009; MAPA, 2021).

The Brazilian government's food purchase programs, in the three governmental spheres, known as institutional purchases and linked to Law 11.326/2006, brought the obligation that at least 30% of the products must be purchased to feed people in situations of food and nutrition insecurity. It is served by the social assistance network, and students served by the school feeding of the public network must come from family farming [1].

These acquisitions, carried out by the PAA and PNAE programs, are vectors for local development, especially in small municipalities that, by creating a potential market, trigger a process of intersectoral articulation, organization of production, and commercialization of products on a larger scale, generating sustainable and food security effects and transforming local dynamics [2,3]. Added to this, the direct food supply from local family farming contributes to an increase in the supply of healthy foods on Brazilian school menus [4]. This dynamic also challenged farmers to understand and adapt production processes so that they could focus on quality and meet the demands of the growing market, fostering its development [2,3].

However, for the efficient execution of these programs, the interaction of different social agents is necessary: farmers, family farming associations and cooperatives, technical assistance and rural extension bodies, entities linked to the food acquisition from agriculture family (program executing entity), and rural unions [2,5]. In the south of Brazil, the institutional environment promoted a more locally integrated dynamic, and schooling and technical assistance stimulated productive efficiency.

Research has shown that, where there is a presence of farmers' organizations or cooperatives mediating transactions between farmers and public authorities, the implementation of programs is more intensely perceived; that is, with a greater food supply and greater participation of family farming in institutional markets [4,5].

The performance of the PNAE in the food acquisition from family farming in Alto Uruguai Territory, located in the North of Rio Grande do Sul (Brazil), stands out for being above the average of Rio Grande do Sul and Brazil [6]. In the execution of the PNAE in the municipality of Getúlio Vargas/RS, for example, in the period from 2011 to 2017, the execution rates were above the state and national averages and reached 63%, and an average of 48%, of the amount transferred by the union used in the purchase directly from family farming [7].

The results come from the organization and commitment of family farming producers in cooperatives, the involvement of local social organizations, such as Technical Assistance and Rural Extension Professional (Emater)/RS-Ascar, the School Meals Council (CAE), and school nutritionists, as well as the unit's executors. Studies show that cooperatives have the ability to articulate with members and organize the production and marketing of products on a larger scale, allowing greater profitability for family farming producers [7–10].

Given this backdrop, understanding how these programs impact on each reality, together with the social agents involved, considering the pluriactivity of family farming in relation to its degree of integration into the institutional market and the local arrangements that influence the way these policies are implemented, becomes a way to contribute to the debate and the systematization of the results of such actions [11–13]. Considering this, the general objective of the research was to analyze the challenges and potentialities in the socioeconomic dynamics of agriculture through the institutional food market in the area of operation of a family farming cooperative. The research was applied in the context of the area of operation of a family farming cooperative, the only one of its kind in the region, which serves farmers in eight surrounding municipalities in the north of the state of Rio Grande do Sul, Brazil.

## 2. Materials and Methods

We used a qualitative approach and case study as a research strategy. The research started from a broad problem, based on a justification at the national level, to study a

regional reality. The case analyzed included the area of operation of Coopraf (Regional Cooperative of Family Agriculture of Getúlio Vargas), headquartered in Getúlio Vargas/RS, contemplating 39 family production units with cooperative members, which belong to 8 surrounding counties. In this region, family farming is common, but only a few producers are part of the institutional market.

The research instrument used for data collection was the semistructured interview, which had as participants family farmers based in Getúlio Vargas (north of Rio Grande do Sul/Brazil) who sell food in the institutional market, the president of the cooperative, the purchasing coordinator of an executing entity of the PAA and PNAE programs, and a technician from Emater/Ascar.

All productive activities carried out in the cooperative's area of action were contemplated, regardless of the agricultural model (crop or animals), with the aim of raising the challenges and potentialities of the institutional market: dairy (1), bakery (3), horticultural products (5), forestry (1), poultry farming (1), sausages (1), hydroponics (1), beekeeping (1), and sugarcane derivatives (1). The interviewees' profile (Table 1) was coded: for family farming producers, codes from E1 to E15 were used; for the other social agents interviewed, they ranged from E16 to E18. The response saturation point was reached in fifteen farmers interviewed.

**Table 1.** Profile of respondents.

| Respondent | Age | Gender | Number of Children | Number of Family Members in the Activity | Intention of Family Succession | Property Size Own Leased (Hectare) | | City | Production/Sector |
|---|---|---|---|---|---|---|---|---|---|
| | | | | | | Own | Leased | | |
| E1 | 47 | F | 2 | 4 | Yes | 30 | 17 | Sertão | Bakery industry in general |
| E2 | 59 | M | 1 | 2 | No | 7 | 0 | Sertão | Honey derivatives agroindustry |
| E3 | 36 | M | 1 | 2 | No | 10.1 | 0 | Estação | Horticultural industry |
| E4 | 48 | F | 2 | 2 | No | 23.7 | 0 | Ipiranga do Sul | Horticultural industry |
| E5 | 50 | M | 0 | 2 | No | 8 | 0 | Floriano Peixoto | Agroindustry of sugar cane derivatives and preserves |
| E6 | 43 | M | 2 | - | No | - | - | Getúlio Vargas | Sausages agroindustry (pork) |
| E7 | 59 | M | 1 | 3 | Yes | 18.5 | 0 | Getúlio Vargas | Horticultural industry |
| E8 | 56 | F | 2 | 2 | No | 12 | 0 | Getúlio Vargas | Bakery industry in general |
| E9 | 51 | F | 2 | 3 | Yes | 15 | 0 | Erebango | Silviculture |
| E10 | 49 | F | 2 | 2 | No | - | - | Erebango | Bakery industry in general |
| E11 | 48 | M | 1 | 2 | No | 2.5 | 0 | Getúlio Vargas | Horticultural industry |
| E12 | 70 | M | 2 | 3 | Yes | 27.5 | 0 | Getúlio Vargas | Egg production |
| E13 | 27 | M | 0 | 2 | No | 0 | 2 | Sertão | Hydroponic horticulture |
| E14 | 18 50 | F M | 1 2 | 4 | Successor Yes | 6.3 | 0 | Getúlio Vargas | Cassava and horticultural industry |
| E15 | 64 | M | 2 | 3 | Yes | 17 | 0 | Áurea | Dairy agroindustry |
| E16 | 32 | M | | | | President of the Cooperative | | | |
| E17 | 49 | M | | | | Executing Entity Purchasing Coordinator | | | |
| E18 | 48 | M | | | | Emater Technician | | | |

The research instrument used for data collection was the semistructured interview script whose participants were family farmers based in the city of Getúlio Vargas (north of Rio Grande do Sul/Brazil), who sell food in the institutional market, the president of the cooperative, the purchasing coordinator of an executing entity of the PAA and PNAE programs, and the technician from Emater/Ascar.

We considered four semistructured interview scripts (adapted from [2,14]) containing thirteen open questions appropriate to the respective respondent social agents and a script for taking notes of direct observations. The interviews followed a conversational mode [15], which enabled bidirectional interaction and allowed the researchers to understand the reality of the participants.

The research procedures constituted four stages: the bibliographical research; the document analysis stage, such as administrative papers of the entity executing the programs; public tender processes, laws, and decrees that regulate this market; the stage of interviews, recorded with authorization, which took place online in September 2020 via the Zoom platform and in person, conducted between 10 and 28 October 2020, observing the COVID-19 period of social distancing, controlled by the government of the state of Rio Grande do Sul. At that time, we carried out the stage of direct observations in the production units, which allowed us to describe reality in a systematic way.

For data analysis, we transcribed the interviews, which generated 179 pages, using the denaturalized transcription technique [16], demonstrating a quality criterion in qualitative research [17]. To examine the information obtained, we carried out a content analysis by categories defined *a posteriori*, considering: economic, social, environmental, food security, and family succession.

Data analysis consisted of breaking down the text into record units (RUs). To do so, first, we manually identified, coded, and grouped the challenges (41 RUs) and potentialities (37 RUs). For the categorization of RUs, we used the semantic criterion (thematic categories) [18]. The results were organized and ordered from the highest number of repetitions (enumeration) to the lowest, and classified according to the five categories.

*Study Object*

Coopraf, headquartered in the city of Getúlio Vargas/RS, aims to organize the marketing of family farming products, a common category in the region. In 2010, it was founded by small rural producers who sought insertion in the regional institutional market due to the existence of institutions (executing entities of the PAA and PNAE programs) in the state with the capacity to consume all the production of the category, highlighting the federal, state and municipal public schools, armed forces (armies), hospitals, and prisons. At the time the research was carried out, the cooperative had 39 cooperative members registered with the DAP (Declaration of Aptitude for Pronaf), recognized by Ministry of Agriculture, Livestock, and Supply (MAPA) as suitable for this sales channel.

The size of the land area exploited by family farmers must be less than four fiscal modules according to Brazilian legislation. A fiscal module is a unit of measurement defined in hectares, with an area previously defined by the National Institute of Colonization and Agrarian Reform (INCRA), and which varies according to each municipality. In all municipalities covered by the survey, each fiscal module has 20 hectares of land.

Its production units are located in eight surrounding municipalities: Áurea (1 property), Charrua (2), Erebango (8), Estação (4), Floriano Peixoto (2), Getúlio Vargas (17), Ipiranga do Sul (1), and Sertão (4) [19] (Figure 1). Among the crops and products of greatest interest are beef, pork, and chicken; eggs; fruits, such as apples and bananas; organic bean and rice cereals; corn and wheat flours; horticultural products, such as cassava, potatoes, tomatoes, onions, garlic, cauliflower, broccoli, seasonings, and salads; products from the bakery industry, which are different types of pasta and biscuits; dairy products, such as cheese and milk.

The cooperative supplies food products to 38 schools: 1 federal institute, 18 state educational establishments, 3 indigenous schools, and 16 municipal schools, demonstrating the broad effectiveness of participation in the PNAE program. On the other hand, in the same scenario, the PAA program, although a tool of similar importance, in the form of institutional purchase, is insignificant in sales to the cooperative, summing up to just a single sales contract for the state prison of Getúlio Vargas [19].

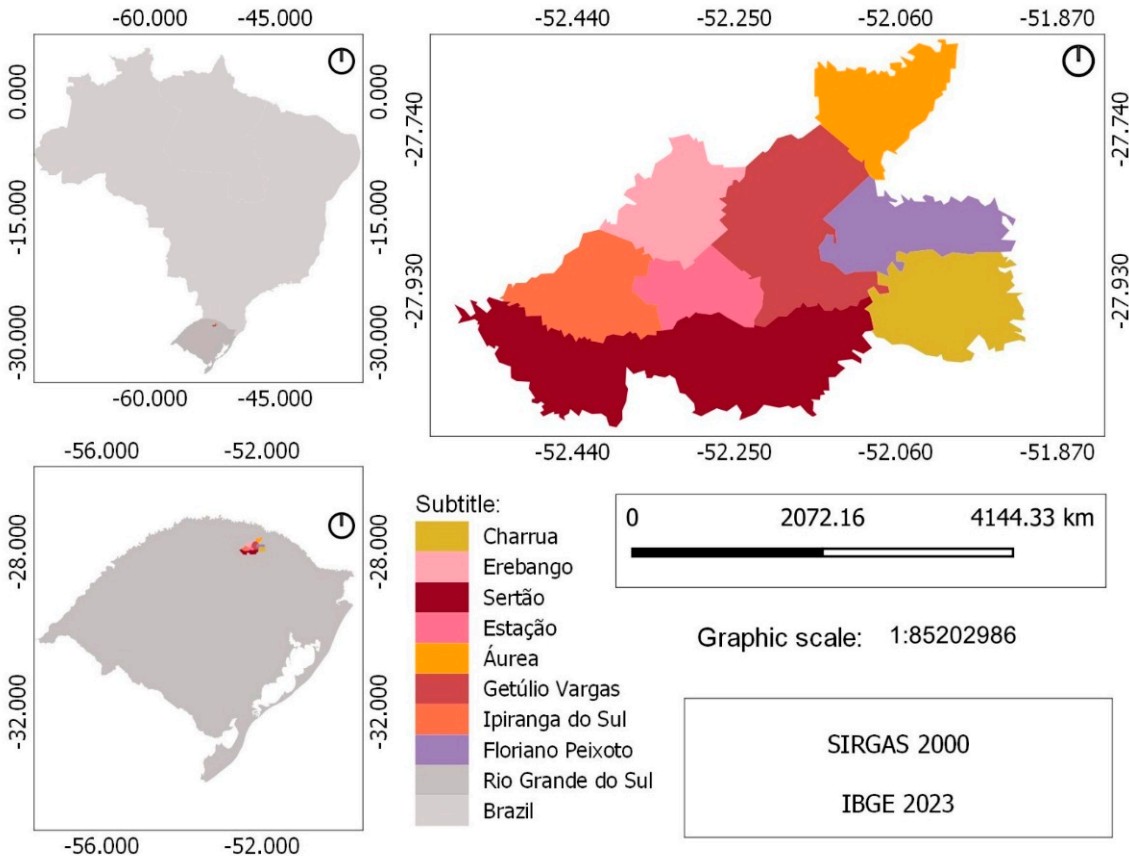

**Figure 1.** Location of cities that include family farmers participating in the research.

PNAE and PAA are federal programs carried out in partnership with the states and municipalities. In the state of Rio Grande do Sul, both programs are operated by the State Department of Education and by municipal governments. Approximately 1.3 million meals are offered per day to 800,000 students in RS (State Department of Education/RS, 2022 [20]).

The transfer of PNAE resources is carried out in 10 monthly installments, from February to November of each year, to cover 200 school days, depending on the number of students enrolled in each education network and value ranges. The average income for the family farmer can be up to 40 thousand Brazilian reals (BRL) per family DAP/year per executing entity of the PAA and PNAE programs.

## 3. Result and Discussion

Our data point out 41 challenges and 37 potentialities for the role of family farming in the institutional food market (Table 2). The largest number of these results are classified in the economic category: 22 challenges and 16 potentialities, followed by the social category with 10 challenges and 10 potentialities. In the family succession category, there were four challenges and six potentialities; in the environmental category, there were three challenges and one potentiality; in the food security category, there were two challenges and four potentialities.

Among the challenges with the greatest emphasis are: labor shortage in the region; excessive bureaucracy; occasional closure of schools due to the coronavirus pandemic; lack of information or disclosure of this expanding market; aging of the rural population; class-based hard work; lack of family succession; high production costs; climate fluctuations; shortage of certain products; sales limits per DAP; outdated prices for certain items and periods; difficulty inserting into traditional communities. Although the support from the cooperative and Emater stands out as a potentiality, some aspects are challenging, mainly due to the need for investment.

**Table 2.** Challenges of the institutional market from the perspective of the social agents participating in the research.

| Recording Unit | Description | Enumeration | Categorization |
|---|---|---|---|
| **Labor shortage** | Shortage of labor and property management. | 31 | Family succession |
| **Excessive bureaucracy** | Delay for legalization; release of traceability agroindustries; electronic invoice. | 28 | Economic |
| **Effects of the pandemic** | There was no delivery; spoiled product; used for animal feed; accumulation in markets and low prices; income decreased due to online classes. | 24 | Economic |
| **Lack of information** | Producers are unaware of the characteristics of the PNAE and PAA programs. | 24 | Social |
| **Lack of succession** | The children prefer to migrate to the city to study and have a better quality of life. | 22 | Family succession |
| **Lagged prices for certain periods and products** | Executing entities should consider the period for preparing the reference value of the products (off-season). | 16 | Economic |
| **Hard work in agriculture** | Hard work, suffering, exhaustion, limitation, and lack of quality of life. | 12 | Social |
| **Need for expansion** | Search for new producers interested in this market, as the demand exists, as well as other forms of acquisition. Introduce new products in the menus of school meals, hospitals, and prisons. | 14 | Economic |
| **Difficulty in integrating traditional communities** | Indigenous peoples and quilombolas do not participate in the institutional market. Prejudice and lack of ethnographic intelligence of managers and of support bodies. | 11 | Social |
| **Conformism and accommodation of people** | Lack of initiative and involvement of the producer himself. | 9 | Economic |
| **High production costs** | Low profit margin. | 9 | Economic |
| **Aging of the rural population** | For many of those interviewed, the family already boils down to the couple, over 50 years old. | 8 | Family succession |
| **Climatic fluctuations** | Loss in production due to climatic factors. | 8 | Environmental |
| **Low participation** | Lack of interest in participating in associations and unions. | 8 | Social |

**Table 2.** *Cont.*

| Recording Unit | Description | Enumeration | Categorization |
|---|---|---|---|
| **Shortage of products** | Little offer of products, such as fish, organic, dairy, eggs, and meat. | 7 | Economic |
| **Cooperative support** | The cooperative is unable to provide technical assistance, logistics, and supply of inputs. | 7 | Economic |
| **Sale limitation** | The sale value per DAP (year) is low. | 7 | Economic |
| **Barriers to organic products** | Lack of plant barrier and lack of stimulus for organic production. | 7 | Environmental |
| **Competition between producers** | Competition between cooperative members from different municipalities. | 6 | Economic |
| **Lack of information exchange** | The lack of union is highlighted by some of the interviewees. | 6 | Social |
| **Need for delivery** | The cooperative did not participate in virtual fairs to reallocate products. | 5 | Economic |
| **Lack of management training** | Need for better planning of rural enterprises. | 5 | Economic |
| **Food waste** | Waste of surplus production due to the greater perishability of some foods. | 5 | Food security |
| **Low performance of Emater** | Teams lagging behind a fundamental support entity. | 4 | Social |
| **Low percentages** | Encourage acquisition of AF beyond the 30% provided by law. | 3 | Economic |
| **Low investment** | Need to invest in irrigation and equipment. | 3 | Economic |
| **Lack of research** | Need for quantitative surveys of supply and demand for products and of new applicants. | 3 | Social |
| **Need to publicize the programs** | Lack of dissemination of programs for producers and consumer beneficiaries, as well as product presentation and dissemination of the "gaucho" flavor seal. | 3 | Social |
| **Lack of leadership** | Lack of leadership, inexperience, and insecurity of young people. | 3 | Social |
| **Lack of equipment** | Some producers do not have a cold room to store products (basic item). | 2 | Economic |
| **Delays in payments** | For specific reasons, there were delays in payments from the cooperative to the producers. | 2 | Economic |

**Table 2.** *Cont.*

| Recording Unit | Description | Enumeration | Categorization |
|---|---|---|---|
| **Use of agricultural pesticides** | There is a need to use chemical treatment, fertilization, insecticides, and fungicides. | 2 | Environmental |
| **Logistics** | Difficulty when the quantity of product to be delivered is simple and with several delivery points. | 2 | Economic |
| **Public investment** | Political crisis; uncertainties regarding budget availability; budget cuts; delays in releasing the budget, mainly in the PAA. | 2 | Economic |
| **Discontinuity of programs** | Adherence to the PAA institutional purchase is embryonic. In other PAA modalities, the price is not compensatory. | 2 | Economic |
| **Acceptance of products** | The culture of the local population made it difficult to accept hydroponic products at the beginning of activities. | 1 | Social |
| **Patriarchy** | Difficulty for the older people to accept changes in the way of production and to accept ideas from the children. | 1 | Family succession |
| **Unfair competition** | Legalized producers have a higher operating cost. | 1 | Economic |
| **Seasonality** | The dairy producer with daily production needs another sales channel during the school holidays. | 1 | Economic |
| **Adequacy of products** | Family farming needs to adapt its products according to the needs of the applicant. | 1 | Food security |
| **Lack of supervision** | Need for inspection on the execution of the programs. | 1 | Economic |

**Frequency of categories:**
Economic: 56.10%; Social: 26.83%; Family succession: 7.32%; Environmental: 7.32%; Food security: 2.44%.

Note. The term enumeration refers to the number of repetitions; that is, the regularity of the recording unit in the course of the interviewees' statements.

The lack of labor in family farming was the challenge addressed most frequently, and this is due, in part, to the fact that families are increasingly smaller, as well as the departure of children from rural areas to cities. These results, classified in the family succession category with the lack of family succession in the productive units and the aging of this population, are worrying factors in this region, as well as being themes explored by numerous researchers [21].

In some production units, it is unlikely that there will be family succession, as some producers are 50 to 70 years old; that is, an aging rural population, which is in line with other studies. In a study in the western region of Santa Catarina, it was observed that most family farms do not have children interested in the succession, while others consider

the lack of skilled labor and generational renewal in agriculture as the main obstacles to increasing production capacity and qualifying production [22].

In family farming, the ideal succession process is when the children join the work on the property from an early age and, little by little, take on tasks of greater importance, reaching adolescence and mastering the production and management methods observed during life. However, young people living in rural areas do not always want to continue the work developed by their parents, even with programs and policies that help them stay in the countryside or when there is encouragement from parents [22].

The aging of the rural population is a relevant topic from an academic and public policy point of view, being a reality in most developed and developing countries [23,24]. Data from the 2017 Agricultural Census show that the rate of young people in the countryside is reducing, while the rural population in Brazil is getting older; this problem is one of the challenges for the maintenance and strengthening of family farming [25], and also impacts on the greater inclusion of family farming in institutional markets, such as the PNAE and PAA [7,26].

Another challenge that stood out in the research is the excess of bureaucracy, observed mainly by farmers, and which is related to issues such as permits and licenses for agroindustries. For managers of executing entities, cooperatives, and Emater technicians and extension workers, even after more than a decade of the operationalization of the programs, it is essential to seek knowledge and update the legislation and procedures of the institutional market. In this sense, the opinion of different professionals from different institutions is essential to understand the difficulties, as well as learning about their duties, as the success of institutional purchasing depends on the involvement and knowledge of different technical areas.

Similar challenges were reported in a study carried out in the state of Paraná, highlighting bureaucracy, health requirements, and the lack of technical assistance in certain municipalities [27]. This issue was also highlighted in studies in the states of Rio Grande do Sul, Santa Catarina, and Minas Gerais [28,29]. The studies indicate that producers need support and technical advice, but public rural extension in certain municipalities has outdated teams. Limiting factors, such as the irregularity of the execution flows of the PAA program, bureaucratic aspects, in addition to the commercialization, limit values [29].

The effects of the suspension of face-to-face classes due to social isolation to avoid the contamination of the coronavirus had a major impact on the period for family farming. With the closure of schools and the establishment of online classes, the expectation of sales of family farming to the institutional market, mainly via the PNAE, did not materialize, and the losses, especially to vegetable producers in view of the greater perishability of products, were unavoidable. Part of the food ended up being wasted, and even used for animal feed, while the price declined for the producer due to the large supply, as the product destined for the institutional market "school lunch" was relocated to the "traditional" market.

As for the executing entities of the programs, it was noticed that they carried out some acquisition processes, seeking to comply with Law N. 13,987 of 7 April 2020, but in a reduced way. The data demonstrate, in addition to the reflection of the pandemic, the ways in which schools stopped demanding products via the PNAE, and consequently how the producer did not make the sale, as well as the importance and dependence of some producers on this commercialization channel in the region. This result corroborates the findings of [3], as a considerable part of the products intended for school feeding are no longer purchased.

### 3.1. Potentialities of the Institutional Market in the Dynamics of Family Farming

After contextualizing the challenges, in Table 3 we present the results regarding the potentialities from the point of view of the social agents participating in the research. Such results demonstrate that the institutional market in the region, mainly via the PNAE, is an instrument for strengthening family farming, positively impacting the socioeconomic dynamics of these producers. The increase in production, work, and income, the strengthening

of local organizations, the improvement in family nutrition, product quality, and environmental awareness are the highlights of the study in the economic, social, environmental, food security, and family succession categories.

**Table 3.** Potentialities of the institutional market from the perspective of the social agents participating in the research.

| Recording Unit | Description | Enumeration | Categorization |
|---|---|---|---|
| **Income guarantee** | Positive impact on the financial return; diversification of the source of income; allowing the subsistence of the families. | 69 | Economic |
| **Institutional relations** | Partnership, cooperativism, cooperation/intercooperation, trust, and articulation between different social agents. | 51 | Social |
| **Cooperative support** | Execution of the bureaucratic marketing process. | 46 | Social |
| **Support from Emater** | Support, encouragement, training, and documentation. | 45 | Social |
| **Women's involvement** | Professional achievement, empowerment, and the appreciation of women. | 45 | Social |
| **Production diversification** | Food diversification, quality and quantity, inclusion of new products, and production capacity. | 40 | Food security |
| **Continued professionalization** | Continuous learning of Emater managers, technicians, and producers. | 37 | Social |
| **Altruism** | Interest, participation, dedication, involvement, sense of belonging, pioneer spirit, and commitment. | 35 | Social |
| **Production planning** | Production and delivery scheduling. Entrepreneurs. | 32 | Economic |
| **Expanding markets** | Sales potential for other institutions, in addition to schools. | 32 | Economic |
| **Acceptance and standardization** | The products are accepted and valued within the required standards. | 32 | Economic |
| **Nutritional quality of food** | Fresh, healthy food, collected and stored properly. | 29 | Food security |
| **Job opportunity** | With the increase and diversification of production, this market provides opportunities for new jobs. | 26 | Family succession |
| **Pricing** | Price considered satisfactory. Adding value. | 26 | Economic |

**Table 3.** *Cont.*

| Recording Unit | Description | Enumeration | Categorization |
|---|---|---|---|
| **Resilience** | Persistence, challenges, hope, and courage. | 23 | Social |
| **Investments in properties** | Investment in production units with returns from the institutional market and financial credit. | 21 | Economic |
| **Legalization of products** | Legalization makes it possible to open markets, validating products not only for the institutional market. | 20 | Economic |
| **Regular payment** | Payment on time by the card system. | 19 | Economic |
| **Family involvement** | The activities encourage the involvement of all family members. | 18 | Family succession |
| **Incentive to stay in the property** | The institutional market encourages and contributes to staying in the field. | 18 | Family succession |
| **Environmental conscience** | Environmental preservation, cleaner production, environmental license, reuse of waste, correct disposal of waste, optimization of water resources, awareness, and environmental responsibility. | 18 | Environmental |
| **Presence of innovation and technology** | Existence of necessary equipment for activities and search for alternative solutions. | 14 | Economic |
| **Inclusion of young people** | Stimulates and promotes the inclusion and involvement of young people. | 11 | Family succession |
| **Self-consumption of food products** | Most of the food consumed by the families is produced on the property. | 11 | Food security |
| **"Positive" effects of the pandemic** | Some activities reached the delivery market and other sales channels, with values higher than the PNAE/PAA. | 11 | Economic |
| **Continuity and expansion PAA/PNAE** | Importance of programs for family farming, ensuring income and permanence on properties. | 10 | Family succession |
| **Organic products** | Satisfactory demand and prices, being supplied by cooperatives from other regions. | 10 | Economic |
| **Return to family farming** | Producers who had left agriculture to work in the city and returned to the countryside. | 8 | Family succession |

**Table 3.** *Cont.*

| Recording Unit | Description | Enumeration | Categorization |
|---|---|---|---|
| **Local development** | Driving the strengthening of family farming to guarantee food production. | 8 | Food security |
| **Favorable logistics** | Producers deliver to schools, which provides another sales channel and direct sales to consumers. | 7 | Economic |
| **Constructive examples, models** | Importance of positive examples and models of success for family farming producers when starting their activities. | 7 | Social |
| **Yield** | The institutional market is considered as the main income of the family. | 3 | Economic |
| **Equipment optimization** | Use of existing equipment on the properties for the demand of the institutional market. | 2 | Economic |
| **Execution beyond 30%** | The IFRS Campus Sertão with execution of 100% of the budget made available by the National Education Development Fund (FNDE). | 2 | Economic |
| **Guaranteed public budget** | The public budget of the PNAE is guaranteed and made available annually. | 2 | Economic |
| **Promotion of education** | The programs provide environmental and food education to consumer beneficiaries. | 2 | Social |
| **Social class change** | Producers boosted by good sales have improved their quality of life. | 2 | Social |

**Frequency of categories:**
Economic: 43.24%; Social: 27.03%; Family succession: 16.22%; Food security: 10.81%; Environmental: 2.70%.

Note. The term enumeration refers to the number of repetitions; that is, the regularity of the recording unit in the course of the interviewees' statements.

The main potentialities are: guaranteed income for producing families; existence of good relationships between institutions, such as executing entities, Emater and cooperatives, and different social agents; visibility of the involvement of women and young people; professionalization, diversification, and planning of production destined for the institutional market; acceptance of products; nutritional quality identified by the consumer/students.

The guarantee of sales and income emerges as the main indicator of the potential of the institutional market for family farming. Respondents were unanimous in stating that this market is an income supplement for the maintenance of families in the countryside and subsistence. Many respondents considered it the family's main source of income, as well as demonstrating a preference for this sales channel to traditional markets. According to statements E14 and E16, in the "traditional market", payment can occur irregularly, as well as the product not sold by the establishment being returned to the producer, a situation that does not occur in the institutional market.

With the results presented, we support the findings of several authors [2–4,8]. The guarantee of commercialization means strengthening the autonomy of family farming producers [30], a factor that encourages the continuity and permanence of families in the rural area. According to interviewees, job opportunities in rural areas have improved as production has increased and diversified, increasing the degree of family involvement in production, marketing, and the organization of the production unit. The inclusion of young people in these activities (E1, E7, E9, E13, E14, E15) is noticed, in which the children are active and even manage the units, as well as the involvement of women, especially in agroindustries, which increases the family income and contributes to improving the quality of life of these families. In this context, the openness of parents, delegating and assigning essential tasks in the activities of the production unit, demonstrates their interest in their children continuing to work in agriculture.

The main restrictions that small farmers faced is market imperfections [21]. They have showed that family farming production was based in two polar cases in the northeast and south regions. In the northeast region, the federal government has established public policies for production growth in top–down instruments by a lack of local integration. In the south region, the local leadership, farmer associations, and state-level authorities were very strong in designing specific policies to promote agriculture. By the same perspective, the productive technical efficiency was estimated by a spatial stochastic frontier model [31]. The results have revealed that the south region presented better performance than the northeast region. The institutional framework behind the production is one step ahead in the south than in the northeast. Cooperativism was important to create a good environment for family farming production in the south. Both studies [21,31] understood that small farms in the northeast depend heavily on income-transfer policies to survive.

There are cases of the returning of people and/or young people who had already left rural areas for urban centers (E8, E9, E13, and E14). This resulting reinsertion of individuals into rural areas creates the necessary conditions for family succession in rural companies. The results corroborate the study carried out in the city of Passo Fundo/RS by Lopes and Basso (2021), which indicates that programs such as the PNAE contribute to staying in rural areas through the generation of work and income. Financial income is one of the main incentives for generational succession in family farming [9,29].

The research carried out by in the MG region [2] showed that the programs have stimulated functional reorganization among family members and provided financial and personal autonomy to young people and women. This insertion and the involvement of young people inhibits migration, the rural exodus, and the emptying of the countryside. The results presented corroborate other studies [2,8], which found an increase in production, work and income, the diversification of production, the strengthening of local associative organizations, and the improvement in family food and product quality. Research shows that the programs had a positive impact on the socioeconomic dynamics of family farming in the MG region [3], just as this execution requires the interaction of several social agents, including family farming producers, cooperatives, technical assistance, and rural extension bodies [2,5].

Another result obtained is that there is planning, organization, and the diversification of production to meet demand. The standardization and legalization of products, although a bureaucratic process was highlighted as one of the main challenges, is seen as an opportunity to leverage sales. Important potentialities are also perceived in the political–social relations of the social agents involved. The implementation of these policies in the region takes place due to the commitment and articulation of several actors, highlighting the producers' organizations (cooperatives), the technical assistance, and rural extension agencies, and the different sectors of the agencies or entities executing the food-acquisition programs.

As the effectiveness of the programs and the ability to serve the institutional market gains capillarity with the insertion and involvement of diverse public agents and social actors from civil society, the importance of cooperatives as a strategy for organizing production and inserting family farming into institutional markets is also highlighted [2,6].

The work of Emater was identified as relevant for most respondents, working both in training and organizing producers, and in offering fresh and minimally processed foods, as well as technical monitoring. This is in addition to highlighting the articulation with the entities executing the programs, which made it possible to meet a demand already installed in this market. In the case of executing entities, Emater developed support work and technical assistance, in the sense of specific legislation, assisting in the process of public tenders, mainly at the beginning of hiring.

It was found, as evidenced [8,28], that public technical assistance plays a relevant role both in training for the management of productive processes and in the elaboration of projects for institutional markets, as well as in encouraging the formation of cooperatives and associations, being essential for a systematic and continuous follow-up. From the case study, the presence of institutional markets and the particularities and diversity of family farming was identified, which requires specific policies for the context of support to the sector. This diversity means that an economically viable production unit can vary according to the region, production strategy, level of market integration, family structure, access to inputs, technology, and infrastructure.

### 3.2. Political Implications

The results highlight challenging questions and imply a set of actions to contribute with guidelines to promote instruments of agricultural policy and of food and nutritional security in the country, which can be used in other regions as well as other countries. In the context of bureaucracy, especially with regard to family agroindustries, the representative entities of family agriculture need to seek legal advice for the preparation of proposals for changing legislation upon consideration by the corresponding legislative sphere. Lagged prices in certain periods and products: the public entity must be warned about the influence of supply and the fluctuation of the price of products during the productive cycle, with the adequate application of public policies that perform due price regulation.

The need for expansion, the shortage of certain products, and the need to adapt products: associative and/or cooperative entities must act in line with the public administration, in municipal, state, and federal areas, encouraging the expansion of family production, and through investments in productive means, storage, transport, and commercialization, thereby allowing more people to be contemplated. Studies with entities linked to research, such as the Brazilian Agricultural Research Corporation (Embrapa), are important to verify the needs of a given region, stimulating production, as well as assessing the potential and characteristics of each production unit. Moreover, in light of market research, promoting the adequacy of products to the local and regional reality, combined with the development of projects that are viable, according to the suitability of each property.

It is necessary to raise the supply and demand for products in the region, as well as new entities that can acquire from family farming, new products that can be produced, and optimizing existing space and equipment. For the cooperative's need for support, high costs, and lack of equipment, city halls, through their secretary of agriculture, can raise funds that produce feasible projects related to the topic, complementing the search for municipal, state, and federal lines of credit, as well as public and private financial institutions.

The establishment of partnerships with teaching institutions that can influence the cost reduction can be cited by training producers and providing continuous monitoring in carrying out daily activities, resolving doubts, and encouraging the inclusion of technical methods and procedures, which result in increased profits, feedback in the system, and motivating producers to continue in their activities and stay in the rural areas.

Another point to consider is the public budget, the discontinuity of programs, and the lack of supervision and investments in the sector. The mobilization of family farming producers through their cooperatives and other associative entities with the municipalities are necessary measures to emphasize with political representatives, especially at the federal level, for updates to the legislation and the effective execution of public policies, emphasizing the importance of supervising the complete execution of programs.

The organization and implementation of an observatory by an educational institution becomes opportune in order to monitor the processes carried out in the region, the releasing of budgets, the participation of family farming, the demand and availability of products, and the needs of producers and executing entities. There is a lack of information and exchange of experiences, and a need to publicize the programs. Moreover, the concerns of the cooperative, the municipality, and other entities, in addition to Emater, that can contribute to greater dissemination of the programs through partnerships with public educational institutions, especially those linked to rural activities, thus clarifying family farming producers regarding to the advantages of participating in government programs for the sale of their products.

Another point is the implementation of measures, such as meetings, agronomy field days, or other actions of a similar nature, for sharing information and experiences, as well as the diffusion of new techniques, the presentation of innovations in terms of equipment and tools, and the broadening of the horizons of knowledge of family farming.

Regarding the lack of research, the increasing of studies in the area with effective participation, in addition to cooperatives, and also of Emater and other associative entities, such as rural unions and city halls, implementing activities in agreement with Embrapa, other public agencies, and the private sector, all aiming to carry out research with the technical and timely application of information, experiences, and data produced, thus contributing to the improvement of production, as well as to knowing the needs and potential of the region.

With regard to the lack of labor and family succession and the aging of the rural population, there is a need for municipalities to implement more public policies to mitigate the difficulties of family farming; among them, the improvement of the telephony and internet signal, which are fundamental for the exchange of experiences, electronic transactions both in the purchase of devices and equipment, and in the sale of products through virtual platforms and, more recently, payment and receipt by electronic means.

## 4. Conclusions

The institutional market is an important strategy that directly contributes to expanding opportunities, considered a guarantee of sales and income, creating local networks for more sustainable production and supply, promoting financial security for producers in the region, in addition to providing opportunities for families to remain in the countryside.

For the continuity of the positive effects, and to mitigate the existing difficulties, there is a priority need for structural investments in complementary public policies, such as technical assistance and rural extension at the local and regional levels, assistance with health inspection, infrastructure and management, support from city halls through the secretariats, engagement of the technicians involved to establish the conditions for inclusive rural development, in addition to the wide dissemination of the positive aspects of the programs to beneficiary suppliers and consumers.

The identified potentialities demonstrate that the institutional market contributes to strengthening the pluriactivity of family agriculture, being a vector for sustainable rural development, mainly in small Brazilian municipalities. In addition, it favors the reduction of socioeconomic inequalities and helps in the advancement of individuals residing in areas with a higher incidence of social vulnerability in rural areas.

This research can contribute to discussions on public policies aimed at family farming, highlighting the relevance of the institutional market via the PAA and PNAE programs, and signaling to public managers to know of such policies. In this regard, instigating the purchase beyond the minimum percentages provided for by law.

As research limitations, we consider the limited number of social agents involved in the interviewed purchasing process, such as nutritionists, school feeding councils, and technicians or rural extension agents from different municipalities. Quantitative research on the institutional market is suggested through the PNAE and PAA programs, involving different cooperatives, a study to map producers, where they are, what and how much

they can produce, as well as to raise the possible executing entities of the programs and the capacity absorption of this production.

**Author Contributions:** Conceptualization, E.R.F.C. and G.d.V.M.; Validation, G.d.V.M.; Formal analysis, E.R.F.C., G.d.V.M. and L.D.M.; Data curation, E.R.F.C. and L.D.M.; Writing—original draft, E.R.F.C. and L.D.M.; Writing—review & editing, E.R.F.C., G.d.V.M., L.D.M., C.R.F., R.B., A.N., C.P.S. and J.E.R.V.-F.; Visualization, L.D.M.; Supervision, G.d.V.M.; Project administration, G.d.V.M.; Funding acquisition, C.P.S. All authors have read and agreed to the published version of the manuscript.

**Funding:** This research received no external funding.

**Institutional Review Board Statement:** Not applicable.

**Informed Consent Statement:** Informed consent was obtained from all subjects involved in the study.

**Data Availability Statement:** Not applicable.

**Acknowledgments:** The authors are grateful for the financial support from the Fundação Meridional (Brazil) and the Federal University of Mato Grosso do Sul (UFMS, Brazil). The authors also thank MSc. Wellington Alvim da Cunha (Federal University of Viçosa, UFV, Brazil), for his contribution to the research insights.

**Conflicts of Interest:** The authors declare no conflict of interest.

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
