# Peer review of "Challenges and Potentialities of Sustainability in the Institutional Food Market of Family Farming"

_sustainability, doi:10.3390/su152215796_

Round 1
Reviewer 1 Report
The manuscript must be deep revised for fluidity and to improve the English language usage. Results must be discussed to explain the results and not only to compare the literature. Applicability must be mentioned.
I have provided my concerns below:
2. Add a statistical results in abstract and highlight the salient findings
3. Keywords: Avoid the words used in the title
4. The introduction section requires more detail to justify the novelty of the work. The authors should consider providing a more thorough literature review to demonstrate how their work builds upon previous studies in the field. Additionally, the authors should clearly state the research gap that their study aims to fill, and explain why this gap is important to address. By providing a clearer justification for the novelty of the work, the authors will help to engage the reader and increase the impact of their study.
5. The logic of the experiment needs to be strengthened throughout the manuscript. The authors should ensure that their methods are described in sufficient detail to allow readers to understand the rationale for each step of the experiment.
6. Additionally, the authors should consider providing a clear and concise summary of the experimental design in the introduction or methods, so that readers have a clear understanding of the purpose of the experiment
7. Furthermore, the authors should ensure that the results are presented in a logical and coherent manner, with clear connections made between the results and the research questions or hypotheses. Finally, the authors should clearly state the implications of their findings, and discuss how their work contributes to the field.
Results and Discussion: The authors should discuss the results in a proper manner. rather than discussing the data presented in tables and figures, authors should provide a proper disucssion with detailed reasoning. The disucssion should be in detail with propert studies
The conclusion should be rewritten.
8. Update the old reference published before 2017 in Introduction and R&D Sections
Please carefully revise the manuscript for the appropriate use of definite and indefinite articles (the, and a/an).
Author Response
Dear Editor
Thank you for taking the time to manage this editorial process and for sending us review comments of revision. We have revised our paper accommodating all the comments. We provide below the point by point reply to the comments.
We reinforce that we are happy with the unanimous recommendation of the reviewer to recommend the manuscript for publication. In this context, the corrections requested were carried out in detail, thus meeting the reviewer' suggestions.
Editor and Reviewer comments:    
First of all, we want to thank the reviewer for the evaluation of our manuscript. We have revised the manuscript according to the anonymous reviewer’ comments. The comments and their replies are shown below. We hope you will consider our manuscript for publication in your esteemed journal. The manuscript is original, no part of the manuscript has been published before, nor is any part of it under consideration for publication at another journal. We convey our immense thanks for dedicating your time to this evaluation. For, without doubt, it helped to improve the quality of this manuscript. Thank you.

Reviewer 2 Report
Peer-review report of the article (sustainability-2587777)
The manuscript entitled “Challenges and potentialities of sustainability in the institutional food market of family farming” is a good article based on an excellent idea submitted for publication in the journal “Sustainability.”
However, this manuscript has some serious flaws that must be addressed before accepting the article.
As the article aims to describe the challenges and potentials of family farming, it is crucial to address some critical questions.
This article is about family farming; however, the authors still need to incorporate crucial information about family farming, such as the prevalence of family farming in the study area compared to other farming types.
What is the average holding size of the family farms?
What are the crops and commodities of focus of most family farms?
How are these different from other types of farming?
What is the possible average income of those farms?
Is family farming common in the study area, or is it deemed a unique field?
According to the given information in the article, the end users, majorly, are institutional users. Are there enough institutions in the state of study to accommodate the majority of the produce?
There needs to be more information about any collaboration between family and corporate farming, which could be a potential supporter of family farming. There is a mention of “Cooperative support” in the manuscript, among the other challenges. Why is there a failure?
As a matter of family possession, an essential element of family farming, the elder's consent for sending their children off the farm and trying to get better education and facilities is also missing. In other words, whether or not the elders want their children to stay and continue farming.
In this study, the opinion of the officials of different agencies concerning family farming needs to be included.
The differentiation of crops and animals must be included, whether family farmers are exclusively going for crops or animals or there is a merger. Moreover, the information needs to be completed regarding the difficulty of different models of family farming.
The article explains COVID-like situations as a challenge for family farmers; however, more information needs to be present about the frequency of such events.
Very generic information is given about the budget. There is no information regarding loans, subsidies, or any other use of the budget, which are crucial for understating challenges and potentialities.
The authors should elaborate on the major challenges and potentialities in detail to complete the manuscript's purpose. For example, the labour shortage, transportation, food waste, unfair competition, payment hurdles, and other vital aspects.
Moderate English and grammar revision is required.
Author Response

(The authors gave the same response as above.)

Reviewer 3 Report
The authors attempted to use a content analysis approach to analyze the challenges and potentials of realizing the socio-economic dynamics of agriculture through institutional food markets in a family agricultural cooperative's field of operation from five perspectives: economic, social, environmental, food security, and family inheritance. It is interesting and valuable.
1. Agricultural development research papers need to have policy recommendations. Therefore, it is recommended that authors should make appropriate policy recommendations in their articles based on the findings of the study.
2. The limitations of the study and future research trends should be adequately explained.
3. The language needs to be appropriately worded.
The language needs to be appropriately worded.
Author Response

(The authors gave the same response as above.)

Round 2
Reviewer 1 Report
No more comments
Author Response
Dear Reviewer,
Thank you for taking the time to manage this editorial process and for sending us review comments of revision.
We have revised our paper accommodating all the comments. We provide below the point by point reply to the comments.
We reinforce that we are happy with the unanimous recommendation of the reviewer to recommend the manuscript for publication. In this context, the corrections requested were carried out in detail, thus meeting the reviewer' suggestions.
Editor and Reviewer comments:    
First of all, we want to thank the reviewer for the evaluation of our manuscript. We have revised the manuscript according to the anonymous reviewer’ comments. The comments and their replies are shown below. We hope you will consider our manuscript for publication in your esteemed journal. The manuscript is original, no part of the manuscript has been published before, nor is any part of it under consideration for publication at another journal. We convey our immense thanks for dedicating your time to this evaluation. For, without doubt, it helped to improve the quality of this manuscript. Thank you.

Reviewer 2 Report
Peer-review report of the article (sustainability-2587777)
The manuscript entitled “Challenges and potentialities of sustainability in the institutional food market of family farming” is a good article based on an excellent idea submitted for publication in the journal “Sustainability.”
The authors have significantly improved the manuscript; however, various questions remain unanswered.
Is family farming common in the study area, or is it deemed a unique field?
There needs to be more information about any collaboration between family and corporate farming, which could be a potential supporter of family farming. There is a mention of “Cooperative support” in the manuscript, among the other challenges. Why is there a failure?
As a matter of family possession, an essential element of family farming, the elder's consent for sending their children off the farm and trying to get better education and facilities is also missing. In other words, whether or not the elders want their children to stay and continue farming.
Is the opinion of the officials of different agencies concerning family farming needs to be addressed?
The differentiation of crops and animals must be included, whether family farmers are exclusively going for crops or animals or there is a merger. Moreover, the information needs to be completed regarding the difficulty of different models of family farming.
The article explains COVID-like situations as a challenge for family farmers; however, more information needs to be present about the frequency of such events.
Very generic information is given about the budget. There is no information regarding loans, subsidies, or any other use of the budget, which are crucial for understating challenges and potentialities.
The authors should elaborate on the major challenges and potentialities in detail to complete the manuscript's purpose. For example, the labour shortage, transportation, food waste, unfair competition, payment hurdles, and other vital aspects.
While writing response, please add the line numbers to indicate the added/edited text in the manuscript. It will be easy to address the question and the response made.
Author Response
Peer-review report of the article (sustainability-2587777)
Dear Reviewer,
Thank you for taking the time to manage this editorial process and for sending us review comments of revision.
We have revised our paper accommodating all the comments. We provide below the point by point reply to the comments.
We reinforce that we are happy with the unanimous recommendation of the reviewer to recommend the manuscript for publication. In this context, the corrections requested were carried out in detail, thus meeting the reviewer' suggestions.
First of all, we want to thank the reviewer for the evaluation of our manuscript. We have revised the manuscript according to the anonymous reviewer’ comments. The comments and their replies are shown below. We hope you will consider our manuscript for publication in your esteemed journal. The manuscript is original, no part of the manuscript has been published before, nor is any part of it under consideration for publication at another journal. We convey our immense thanks for dedicating your time to this evaluation. For, without doubt, it helped to improve the quality of this manuscript. Thank you.
REVIEWER: 2
The manuscript entitled “Challenges and potentialities of sustainability in the institutional food market of family farming” is a good article based on an excellent idea submitted for publication in the journal “Sustainability.”
The authors have significantly improved the manuscript; however, various questions remain unanswered.
Authors respond: Thank you for your important words in acknowledging the quality of our manuscript. Thank you also for helping us to improve the quality of our manuscript. To this end, our changes in the revised version of the paper are highlighted with revision marks, in order to facilitate the reviewer's access to the requested modifications. Thank you much indeed!
Is family farming common in the study area, or is it deemed a unique field?
Authors respond: Thank you for your important words in acknowledging the quality of our manuscript. We have added more information about what was requested at this point in lines: 102-103 and 110-111.
There needs to be more information about any collaboration between family and corporate farming, which could be a potential supporter of family farming. There is a mention of “Cooperative support” in the manuscript, among the other challenges. Why is there a failure?
Authors respond: Thank you for your important words in acknowledging the quality of our manuscript. In the authors' understanding, this information is provided in subsection 2.1. Study Object, especially between lines 147-160.
In addition, we emphasize that this article does not refer to corporate agriculture. Thus, the profile of interviewees only considered family farming; small farmers who, with the help of Emater, formed a family farming cooperative to have access to the institutional market. Corporate agriculture includes more than four fiscal modules that cover family farming.
As a matter of family possession, an essential element of family farming, the elder's consent for sending their children off the farm and trying to get better education and facilities is also missing. In other words, whether or not the elders want their children to stay and continue farming.
Authors respond: Thank you for your important words in acknowledging the quality of our manuscript. We have added more information about what was requested at this point in lines: 300-302.
Is the opinion of the officials of different agencies concerning family farming needs to be addressed?
Authors respond: Thank you for your important words in acknowledging the quality of our manuscript. We have added more information about what was requested at this point in lines: 238-241.
The differentiation of crops and animals must be included, whether family farmers are exclusively going for crops or animals or there is a merger. Moreover, the information needs to be completed regarding the difficulty of different models of family farming.
Authors respond: Thank you for your important words in acknowledging the quality of our manuscript. We have added more information about what was requested at this point in lines: 109-116. In addition, the authors reinforce that the main objective of this work was to identify the challenges and potentialities of the institutional market, not specifically the different models of family farming. In this case, the cooperative under analysis is formed by qualified family farmers and seeks to meet the varied food demand, be it crops and/or animals.
The article explains COVID-like situations as a challenge for family farmers; however, more information needs to be present about the frequency of such events.
Authors respond: Thank you for your important words in acknowledging the quality of our manuscript. We have added more information about what was requested at this point in lines: 260-262.
Very generic information is given about the budget. There is no information regarding loans, subsidies, or any other use of the budget, which are crucial for understating challenges and potentialities.
Authors respond: Thank you for your important words in acknowledging the quality of our manuscript. We have added more information about what was requested at this point in lines: 183-187.
The authors should elaborate on the major challenges and potentialities in detail to complete the manuscript's purpose. For example, the labour shortage, transportation, food waste, unfair competition, payment hurdles, and other vital aspects.
Authors respond: Thank you for your important words in acknowledging the quality of our manuscript. We have added more information about what was requested at this point in lines: 195-202, 238-241, 260-262, 274-279, 300-302.
While writing response, please add the line numbers to indicate the added/edited text in the manuscript. It will be easy to address the question and the response made.
Authors respond: Thank you for your important words in acknowledging the quality of our manuscript. We followed the advice and added line number to each of the answers.
With our best regards,
The Authors.

Round 3
Reviewer 2 Report
The authors have thoroughly improved the manuscript and addressed the reviewer's comments.
The authors have thoroughly improved the manuscript and addressed the reviewer's comments. The manuscript could be accepted after language and grammar proofreading.